# The Ghosts Navalny Met: Russian YouTube-Sphere in Check

**Yulia Belinskaya**

Department of Communications, University of Vienna, 1090 Vienna, Austria; yulia.belinskaya@univie.ac.at

**Abstract:** The disrupted Russian media ecosystem facilitates the flourishing of political blogging on social media; the political communication expands beyond the over-controlled, institutionalised channels of political interaction. This paper maps the activity on YouTube, incorporating media and communication studies to the analysis of a hybrid political regime to answer the following research question: *What representations of Navalny are available in the Russian YouTube-sphere?* The analysis of a statistically random sample of 366 videos associated with the keyword "Navalny" works in two phases: image type analysis and narrative analysis. Both phases help to identify the traditional institutionalised political actors in the spillovers of political communication trends in the YouTube-sphere. This work not only enriches and updates the current understanding of the Russian political communication ecosystem, but also helps expand the research on contemporary quasi-democratic political scenarios and the communicative strategies of their legitimisation.

**Keywords:** YouTube; hybrid regime; legitimisation; Russian media; Navalny



## 1. Introduction

The institutional political and media ecosystems in Russia have been tightened by regulatory control (Gureeva and Samorodova 2019), pushing political debates beyond their limits. The Russian case is paradigmatic of a global process: political communication overspills the institutionalised channels of political interaction (Nulty et al. 2016) represented by circuits that interface governmental executives, politicians and parties, broadcasting press, and media. YouTube epitomises this extended quasi-public sphere that enables dissent voices with the chance of reaching their audiences (Edgerly et al. 2009). As a part of Google machinery, YouTube falls beyond the reach (Litvinenko 2021) of the direct legislative pressure in Russia to remain a relatively free platform not only as a source for entertainment, but as a space for political communication. This paper gravitates in the space between YouTube's relative freedom and its non-neutrality as a communication platform (Morozov 2012). The euphoric academic approach that has treated YouTube as a chance to "break" the glass ceiling of a controlled public sphere (Edgerly et al. 2009) needs consideration. This paper shows that, despite its corporate "independence", YouTube does not break such a dynamic but rather extends the repressed media ecosystem by allowing governmental intervention through content-management.

This article explores the case of Alexei Navalny as an example of such a political communication overspill in the Youtube-sphere. Navalny started channeling oppositional discourses on YouTube in 2013. His activity at the margins of the Russian political public sphere enriches the political debate while challenging the controlled Russian space of traditional political communication, which has responded with disperse but strategic governmental action consisting of silencing and discrediting him with the help of astroturfers (Belinskaya 2020). Navalny's presence on YouTube helps us understand the limits and features of the Russian extended public sphere.

The paper starts with a state of the art of the Russian political regime and the dilemmas of hybrid or neo-authoritarian states (Petrov et al. 2014) and the subsequent metamorphosis of the media ecosystems (Denisova 2017). A discussion on anti-Americanism and other US-related plots follows, to illustrate the legitimacy strategies of neo-authoritarian states.

The methodology and analysis sections describe the sampling process of 366 videos out of a collection of 7449 videos, and show the 'anti-Navalny' plots often rooted within the overarching elite-driven, anti-American discourses that associate Navalny with US funding and American propaganda strategies.

The paper contributes to existing research by enriching and updating the current understanding of the Russian political communication ecosystem and also expands the research on contemporary quasi-democratic political scenarios and the communicative strategies of their legitimisation.

## 2. Mapping the Russian Political and Media Ecosystems

*2.1. The Russian Political System: This Is Neither a Democratic nor an Authoritarian Arrangement*

The Russian political system is classified as a republican democracy by the Constitution of Russian Federation (). Within the academic discourse, it is described as illiberal (Zakaria 2007), authoritarian (Ambrosio 2016), defective (Merkel 2004), uncertain (Mitchell 2013), hybrid (McMann 2006), neo-authoritarian (Becker 2004), a pseudodemocracy (Diamond 2015), and an informational autocracy (Guriev and Treisman 2019). Since the start of the "third wave" of democratisation in the 1970s (Huntington 1991), there has been no scholarly consensus on what actually constitutes democracy and on which institutions and freedoms signify a fully transitioned democratic regime (Diamond 2015), and Russia seems to fall in between the definitions maze.

The concept of 'defective democracy' (Merkel et al. 2003) prescribes a system that is supposed to "observe the formal procedures of electoral democracy but combine[s] them with autocratic characteristics" (Croissant 2004, p. 158). Three dimensions of democracy should be measured in order to classify it correctly: freedom, equality, and limitation of political power; alongside five democratic institutions—decision making, intermediation, communication, legal guarantees, and implementation (Bogaards 2009). The main preconditions for a defective democracy to transform are "strengthening of both political and civil rights, of stateness, of institutional stability, and/or of rule of law" (Croissant 2004, p. 205). Furthermore, nine factors affect the state of democracy: socio-economic modernisation, distribution of wealth, political culture, historical legacy, the integrity of a state, 'nation-building', design of political institutions, the division of legislative, and executive power (Merkel et al. 2003). A regime can be called democratic if it adopts the following institutes: universal suffrage, fair multiparty elections, and media pluralism (Morlino 2009).

The current regime in Russia is regularly classified as a 'hybrid regime' (Treisman 2011; Liu 2020). That term traditionally describes a transitional form of the political regime from authoritarian (or any other non-democratic regime) towards a democratic one or one that has been born out of the crisis of democracy, therefore possibly acquiring or losing some characteristic traits of democracy (Morlino 2009). These regimes may have adopted the form of electoral democracy, however, in cases of multiparty elections, they could be masking and *legitimising* an autocratic power regime (Diamond 2015). In this context, 'hybrid' also becomes a misleading term, suggesting from its name that the regime combines traits of authoritarian and democratic regimes; however, it is in fact an authoritarian regime, and hybridity means it is transitional (Shulman 2020).

Scholars underline the tendency of hybrid regimes to stabilise and democratise (Morlino 2009). According to Brownlee (2009) "hybrid regimes have better prospects for democratisation than fully closed regimes" (p. 516). According to The Economist Intelligence Unit (2021), the regime is called hybrid if the democracy index is between 4 and 6; whereas, since 2019, Russia scores 3.1 placing within the pure authoritarian regime index. Litvinenko (2021) describes the transition of the Russian regime from anocratic or semi-authoritarian towards authoritarian. Moreover, Morlino (2009) leans towards a certain transition to authoritarianism. Such lack of consensus on the form of the hybrid-ness of the Russian regime makes of the country a "world innovator in nondemocratic practices" (Petrov et al. 2014, p. 2).

Holmes (2015) criticises the authoritarian-democratic polarity of regimes used by scholars to explain the Russian case: "The regime's desire and ability to render its actions opaque and illegible to the public, however, does not make it into an authoritarian regime in the classical sense" (p. 40). The idea that the regime is not only mimicking a democracy but also simulating a dictatorship that does not exist in reality is also voiced by Shulman (2020). Holmes (2015) insists that the elites do perfectly understand the dysfunctionality of the state and the inability of the political leadership to solve the most prominent problems of the country: "This is neither a democratic nor an authoritarian arrangement" (p. 41).

The systematic 'counter-mobilisation' described as a "non-intrusion pact" between the ruling elites and the civil society to keep the status quo is another feature of hybrid regimes. This is characterised by Petrov et al. (2014) as: "so long as Russians see that the state does not significantly intrude into their private life pursuits and is at least minimally delivering actual governance, they are willing to put up with, and adjust to, dysfunction and even decline" (p. 6).

All those authors only briefly touch upon non-governmental institutions and on the role of media in the maintenance and sustainability of democracy and their effects on the deterioration of democracy. Communication scholars, on the other hand, fully engage with the role of media in the facilitation of democracy, with the earliest debate around the educational role of television in electoral processes emerging in the 1950s (Trenaman and McQuail 1961). With the rise of the Internet, the discourses around digital media and their role in democracy have re-emerged.

### 2.2. The Media Landscape: Reporting a Hybrid Regime

The media landscape in Russia is characterised by the consistent capture of media, conveyed in the acquisition of national broadcasters, newspapers, and news agencies (Becker 2014) and accompanied by waves of regulatory restrictions (Gureeva and Samorodova 2019). Amongst these policies, "anti-terrorist" laws have been particularly useful in concealing practices of surveillance and violation of privacy, not only in Russia but all over the world (see Hass 2009; Eroukhmanoff 2016).

Overall, the media play a fundamental role in contributing to the public debate and they eventually granted the legitimacy of these laws (Wæver 2015). Hence, in an environment of state-owned media, the conditions for such debate might have been dramatically adulterated (Walker and Conway 2015).

In front of the radical reshaping of the Russian media landscape, international organisations have expressed their concerns about the freedom of the media in Russia and the consequent democratic health, including Freedom House (2012), Human Rights Watch (2017), and the International Press Institute (IPI 2002). During the recent decades, governmental networks extended to reach the steering wheels of multiple private outlets of the media ecosystem. The demise of private media did not start in 2001 with the first Russian private channel NTV. Indeed, thanks to the "privileged access to political power" (Shinar 2015, p. 588) that the owners of private outlets had (for instance, Russian oligarchs Gusinsky and Berezovsky), the independency of those media are put in question, especially in regards to election campaign coverage in 1996 (Guriev and Rachinsky 2005). Therefore, when the critical and investigative work of NTV during Chechnya War led to its forceful nationalisation (Irisova 2015), that was not the first case. As a result, "by 2005, not a single national television channel was free from state control" (Bacon et al. 2016, p. 76).

The following electoral periods brought new waves of legal 'reformation'. In 2012, a series of amendments and legal changes were implemented: the NGO law amendments (the so-called 'foreign agent' status for organisations); the amendments to the criminal code concerning defamation, high treason, and state secrets; and the bill "On the protection of children from information harmful for their health and development" (called the "blacklists bill"). In 2013, new amendments increased liability for insulting the feelings of believers and established responsibility for promoting non-traditional sexual relations (Bennetts 2017). These policies tightened the media regulatory framework, especially within the

discourse on securitisation against terrorism, after which followed a targeted closure or 'reconstruction' of multiple media outlets and news agencies (RBK, Forbes, TV2, RenTV, Lenta.ru, RiaNovosti, Kommersant, gazeta.ru, Vedomosti). This chain of actions substantially modified Russian media ownership and the ideological landscape and, with it, the quality of the Russian democratic space.

This Russian media environment fits well within a neo-authoritarian or hybrid context—a system in which media pluralism and private ownership are tolerated but political power is used by the media as a weapon to attack political enemies. These regimes also suffer from the bifurcation between the print media and television, with the latter being the most important medium under state control and the print media remaining relatively independent. Such relative independence adds to a few scattered elements of democratic regimes that create the "appearance of democratisation" (Becker 2004, p. 150).

In Russia, state-owned or state-related media do not enjoy independence for anything concerning appointments to editorial positions and editorial policy; while there is no direct censorship, the state intervenes in the control of the media with soft financial mechanisms, such as taxes or subsidies, or through the legal instruments described above. Such tolerated covered financial violence exerted upon oppositional media leads, in the practice, to their own self-censorship. According to Erzikova and Lowrey (2017) local journalism is also fragmented and heavily dependent on governmental subsidies (Erzikova and Lowrey 2020). The system of reliable functionaries as well as directors and editors in the provinces aim to demonstrate loyalty to Kremlin and to receive the financial benefits (ibid.).

Bodrunova (2013) and Petrov et al. (2014) describe this discrepancy between the democratic self-representation of the regime in legal documents and political speeches and the actual autocratic characteristics including the elimination of "political and media opponents, directly controlling regional elections and corrupting the economy" (Denisova 2017, p. 977) as a hybrid regime. The regime in Russia tries to disengage the public through increasing control on media markets, the suppression of independent journalism, and the co-optation of digital media (Petrov et al. 2014).

Neo-authoritarian or hybrid regimes are often characterised by general low trust in media and in political institutions, especially among younger generations. In Russia, this also includes those older than 45, who are usually more active in electoral processes (Levada 2019). About half of younger respondents receive their news from internet sources daily (WVSA 2020). In this sense, social media gained trust recently, and video blogs enjoy new popularity as trustworthy sources of political information; they are watched at least once a week by a third of Russians (two-thirds of younger audiences); similarly, Telegram channels have become important sources of news.

### 2.3. YouTube as an Alternative Television?

Since its start in the mid-2000s, YouTube has evolved into a multi-purpose platform that also serves as a space for political communication (Litvinenko 2021). The evolution of the platform curves from the "amateur media" to a formalised structure of corporate practices with enormous monetisation possibilities (Cunningham et al. 2016). Van Dijck (2013) described YouTube as a new but fully institutionalised member of a media market and not an alternative television.

"Video-sharing", which also includes viewing, commenting, liking, subscribing, and all other possible activities (Van Dijck 2013) rapidly became the means of connection for communities, providing a new quasi-public sphere with low entry barriers which is almost unmoderated and non-discriminatory. The initially considered "euphoric" (Van Dijck 2020) academic discourse around the possibilities of YouTube and other platformised social media capable of empowering ordinary users changed a little later. As noted by Poell and Van Dijck herself (2015), "social media do not simply enable user activity, but very much steer this activity" (p. 528), and as stated by Ha and Yun (2014), most of the YouTube audience are still passive users that do not engage in any activities, rather than occasional ad hoc viewing. Moreover, the recent discourse around YouTube involves disinformation (Hussain

et al. 2018), fake news (Tuters 2021), algorithmic inequalities driven by commercial interests of the tech companies (Burgess and Green 2018), and the responsibility for transmission of public values (Van Dijck 2020).

YouTube, therefore, is not to be addressed as a neutral space of free communication. Clearly, YouTube creates a major interest for advertisers due its structure: YouTube communities connected by certain interests, cultural backgrounds, lifestyles, and by the algorithm (Van Dijck 2020). To foster the advertising revenues, algorithms personalise the experiences of the viewers, creating algorithmic "echo chambers" of very narrowed recommendations and filtered search results (Burgess and Green 2018). Popular uploads and certain actors are more likely to become systematically featured (Rieder et al. 2020), which creates a powerful hierarchical structure and cultural homogeneity. Moreover, the content-removing policies of the platform have also been questioned (see Morozov 2012).

In Russia's suppressed and co-opted media environment, which is consistently trending towards full control over the online realm, YouTube has still been tolerated and presented as a relatively free independent space capable of giving a voice to political dissent. According to Goncharov (2017), the videos with political content remained featured among trending videos. The state of the Russian media system have largely contributed to the overspill of the political campaigning towards YouTube: "The blogosphere becomes a kind of non-institutional digital space for the manifestation of civic agency and political participation, contributing to the formation of political attitudes in society and predetermining vigorous actions of citizens" (Blinova et al. 2019, p. 92).

Navalny and his team succeeded in creating their first YouTube channel in 2013; however, starting from 2017, their YouTube network reached expanding audiences, allowing them to crystallise in actual protests (see Belinskaya 2020). This coincided with YouTube steadily occupying a third place among the most visited online sites in Russia, only behind vk.com and yandex.ru. A recent video published by Navalny in January 2021 about the "palace of Putin" has received 118 million views (state August 2021).

Despite the efforts of the Russian media regulatory waves, the regime seemed to tolerate YouTube. Such exceptionality can be traced along with several reasons:

First, YouTube was often perceived by the Russian regime as a medium for younger viewers, as opposed to television aimed at ageing audiences (Poluekhtova 2012), and only after the viral video investigation published by Navalny that led to the so-called "protests of schoolchildren" in 2017, these younger viewers became part of the oppositional discourse, and YouTube attracted the attention of political analysts and of the Russian elites (Litvinenko 2021).

Second, YouTube was exempted from control because of its capacity to attract an audience wider than the state TV channels in charge of professional high-quality entertainment content (Petrov et al. 2014): "The mass audience of the three major national networks is, in an important sense, the electoral base of the regime" (p. 8).

Third, YouTube is a part of a large American corporation that has enjoyed relative independence from the Russian judiciary system. There are several court decisions justified by "insults on the reputation" prescribing Navalny to delete certain investigations from the channels; however, none of these decisions has been followed. Over the years, there have been several attempts to impose fines on YouTube, Twitter, and Facebook; however, only attempts to fully block Telegram took place on the territory of Russia between 2018 and 2020. In December 2020, the State Duma adopted a law prohibiting censorship of Internet portals. The law allows Roskomnadzor[1] to impose administrative fines or to completely or partially block big social media platforms for "discriminating against materials from Russian media".

Finally, as in the case of Telegram, the phenomenon of YouTube in Russia can be explained through the media dilemma of a hybrid regime.

*2.4. The Three Dilemmas of a Hybrid Regime*

Three dilemmas identify hybrid regimes, and they are clearly visible in the Russian political and media landscape: election dilemma, state dilemma, and media dilemma (Petrov et al. 2014).

The election dilemma:

Free and multiparty elections create political uncertainty for the regime, but at the same time, they assure legitimacy and offer various benefits to the regime by providing "a peaceful mechanism for resolving differences of opinion <...> generate public interest and hence engagement in the policy process and incentivise innovation in policy-making" (Petrov et al. 2014, p. 3). Elections provide an opportunity for the political opposition and critics of the regime to channel the challenges into a peaceful settlement rather than into the organisation of revolts and uprisings. Hybrid regimes tend to solve this dilemma "neither by accepting free and fair elections nor by eliminating elections" (p. 4). In practice, this means that the regime imposes various formal and informal mechanisms of filtering oppositional candidates to lower the political uncertainty starting from "municipal filters" (when the candidates should collect thousands of signatures of voters to prove their eligibility), to widespread election fraud (Robertson 2017). These mechanisms can also include extensive media coverage of candidates from ruling parties, the marginalisation of the opposition, or the introduction of "spoiler" parties that seem independent from the ruling elite and would draw some oppositional votes (Golosov 2015).

The state dilemma:

The dilemma of the state concerns the main political institutions that work as pillars of the democratic model such as the Parliament, but that then, as in the case of modern Russia, lack independence, authority, transparency, and representativity. The dilemma is solved with the implementation of a system of substitutions such as the State Council and the system of "intensive manual control" (Petrov et al. 2014, p. 11), meaning that there is a new opportunity for the intervention at the highest level of leadership into the local affairs, especially during major social shocks and reducing the possibility of an uncertain outcome, or of any local or regional independent gesture.

The media dilemma:

It describes the necessity of the regime of high-quality information that is used to adjust policies or to react rapidly to the changing public inquiries. Media pluralism and high-quality journalism provide low-barrier access to information to the public as well, which, combined with political competition, create sufficient threats for the regime. According to Petrov et al. (2014), the media dilemma in Russia has been solved in the same way as the election dilemma—by retaining media pluralism to a certain degree and allowing some of the oppositional media to operate but demolishing all possible threats. The recent wave of attacks towards independent media outlets, such as digital newspaper Meduza and private TV channel Dojd', which in 2021 received the status of "foreign agent" and lost all the advertising revenue (as well as the law on news aggregators passed in 2020), shows that the tightening of the media pluralism is continuing, and only the media outlets remaining under the direct control of the ruling elites are allowed to operate. The rise of Telegram as a source of "insider news" for both the elites and the public in a framework of a disrupted media environment is a very symptomatic development (Reut 2019).

However, with the rise of the internet and digitalisation, the media dilemma has been updated into the new digital dilemma that is described as dictator's dilemma.

*2.5. A Russian Dictator's Dilemma*

A decade ago, Shirky (2011) described the dictator's dilemma: the access to information of netizens that mobilise thanks to their increased literacy produced by the internet penetration creates problems for authoritarian regimes. Thanks to the networked structure and global penetration of the digital realm and the social media platforms, the digital sphere remains relatively free and safe despite the governmental "efforts to impose control over traditional media" (Denisova 2017, p. 977).

However, Russia has challenged that idea by opening five fronts of action that affect the digital communicative spaces (Rodriguez-Amat and Brantner 2016). Three of those (1, 2, 3) have been identified by Denisova (2017), and the other two (4, 5) are from Belinskaya (2020):
Numbered lists can be added as follows:

1.   The intervention on the ownership of the broadcasting media, ensuring a state control of the most popular Russian channels;
2.   The increase of economic pressure on private corporations;
3.   An architecture of laws that enable the government to prosecute authorship or publication on media outlets under legal pretext;
4.   The recent package of laws that focus on the digital media;
5.   Intervening in the media contents with steered user-generated content: astroturfing.

The vague wording in the new regulatory texts contributes to an ambiguity that opens interpretive space and preemptive capacity to the judiciary and executive powers. With those, they can restrict the media and weaken them and make them vulnerable to state abuse. Following Denisova (2017), self-censorship has thus become the new normal among critical bloggers and discordant media professionals.

The law about news aggregators that came into power in January 2017 forced all news portals to check the "reliability of information" from all sources not registered by Roskomnadzor as media outlets. This immediately meant that these non-registered sources (including the foreign press) were threatened with disappearance in the news aggregates. Such a piece of regulation influenced services such as Yandex.news, the main page of which was opened daily by more than 80% of the Russian internet users (Prokhorushkin 2014).

Aligned with this package, the 'Yarovaya law'[2] impacted the messaging platform Telegram, which was restricted in Russia in April 2018. In June 2020, Roskomnadzor lifted the ban in contradiction with the 2018 court decision setting a grey legal environment and a precedent that points at the glitch between the legal codes. Finally, in 2019, a 'Sovereign Internet Law' was developed and applied.

Astroturfers are the fifth of the fronts used by the government to intervene and alter the public debate. Paid online commentators (also called astroturfers among others by Miller 2016; Zhang et al. 2013) spread (dis)information and foster illusions of mainstream opinions aligned with the state rhetoric. This is a strategy of the government to intervene in the media content with controlled and paid user-generated content. Astroturfers and bots, addressed by Humprecht et al. (2020) as "'undefined' actors, influence the distribution of political information, and contribute to a skewed representation of viewpoints encountered online" (p. 496).

These are five media governance strategies used by Russia and typically by hybrid neo-authoritarian regimes to keep control on the communicative ecosystem that contributed to the ranking of Russia at 156/179 in the liberal democracy report in 2019 (V-Dem 2020).

### 2.6. Legitimisation of the Regime

Hybrid or neo-authoritarian regimes seek legitimisation to keep control in the situation of a crisis of trust. This is, following Bäckstrand et al. (2008) to keep a shared public acceptance as a regulatory regime. Legitimacy, however, is a complex concept that seizes different dimensions: it can be internal and external (Bellamy and Castiglione 2003), structural and behavioural (Murray and Longo 2018), and normative and social (Lindgren and Persson 2010). The normative or external legitimacy refers to a supposedly objective congruence of a certain policy or the regime in general with the universally accepted set of standards, such as the framework of human rights or legality. Social or internal legitimacy involves the acceptance by a particular public based on a set of conventional norms and beliefs (Beetham 2013) or so-called "soft law" (Burgess 2002).

The process of earning legitimacy as "a justification of a behavior" (Reyes 2011, p. 782) is exercised through a 'communicative' or 'discursive' strategy: "these legitimacy claims make use of value-laden language to (re)define and present the institution as a force for a normative good, such as poverty reduction, the protection of human rights, or the

promotion of democracy" (Gronau and Schmidtke 2016, pp. 541–42). At the same time, the legitimisation process not only involves justification and explanation from the side of the regime but also compliance of the public with actions and policies and obedience to the decisions and orders (Reus-Smit 2007). When the discursive action is not sufficient to gain legitimacy among the public, the regime may undertake some institutional change, introduce new procedures and involve more citizens in decision making. However, as discussed above, hybrid regimes employ counter-mobilisation and do not involve the general public in the governance process.

There are various discourses that are commonly used to legitimise authoritarian leadership, the most common of which is an appeal to national security, foreign enemies, or economic stability (Rodríguez-Amat 2015). In the case of China, there are two main strategies that are indicated as legitimising—economic growth and nationalism. "This is to say economic success is not per se a source of regime legitimacy; instead, it has to be framed in ways conducive to positive subjective perceptions of the regime, for example, as competent, efficient, fair" (Holbig and Gilley 2010, p. 400). The nationalistic sentiment also involves anti-Western or anti-Japanese discourses. In the case of Russia, the anti-Western sentiment is also prevalent in the rhetoric of the elites and as underlined by Katzenstein and Keohane (2007), "anti-Americanism can be a potent and useful stand-in for otherwise missing symbols of collective identity" (p. 13).

### 2.7. Anti-Americanism

The Russian elites have repeatedly flagged anti-Americanism discourses, particularly during the period 1999–2008, according to Zimmerman et al. (2020): the political tensions between Russia and the US have recurred, especially during certain peaks, such as, for example, the bombing of Yugoslavia in 1999, the Russo—Georgian War in 2008 or the events in Crimea in 2014 have triggered and exacerbated these narratives.

In general, surveys show that more than half of the Russians have more faith in the American electoral system than in their own, which is clearly different from the attitudes towards American foreign policy (Shlapentokh 2011). Culture, lifestyle, and economic system are also often perceived in a positive or neutral way; however, the American threat against Russian sovereignty is actively repeated by Russian political elites.

Anti-Americanism is treated by this paper as a decentered concept that helps incorporate critical theoretically relevant perspectives to the literature that overly gravitates with the reference of the "American way" of dealing with media and communication. The concept of anti-Americanism is multidimensional, and it involves cognitive (a systematic prejudice affecting the interpretation of available facts), emotional (serves to intensify the assessments), and normative components (when the assessments shape the behaviour). The feeling also may be described as assessments of a particular individual or as collectively shared beliefs aimed at American society and American lifestyle, or at American foreign policy, whatever they might specifically mean.

Zimmerman et al. (2020) reported a steadily growing trend of anti-Americanism for both elites and the public: starting from the early 2000s, more than half of the respondents perceived the US as a threat to national security, and even after a short decline in 2012, the overall public mood of anti-Americanism was still higher than at the beginning of the 1990s. The most negative trends toward the United States among the Russian public were detected after the invasion of Iraq (2004) and during the Russo—Georgian War (2008). Werning Rivera and Bryan (2019) also point at a Russian anti-Americanism peak in 2016.

There are several explanations for the phenomenon of anti-Americanism in Russia. Zimmerman et al. (2020) discussed the theory of "ressentiment"—the frustration of the elites and the population in the democratic reforms of the early 1990s which has reflected in attitudes towards the United States as a state that served as a model for such reforms: "if the borrowing of this experience does not lead to the achievement of the set goals, then the population develops a kind of inferiority complex, which develops into aggressive nationalism and hostility to the state that was previously a model" (p. 41). However,

anti-Americanism continued to rise despite the obvious economic growth at the beginning of the 21st century. The authors thus discussed the hypothesis that after the beginning of the 2000s "a positive assessment of the current political and economic course pursued by the government has become a factor fueling anti-Americanism" (Zimmerman et al. 2020, p. 43).

Whatever the case, it is particularly relevant to note that both Zimmerman et al. (2020) and Shlapentokh (2011) insist that such anti-Americanism feeling began earlier and is more widespread among the elites than among the public: "Anti-Americanism in Russia, as well as in most other countries, does not come from below, from the general Russian population, but rather from above, from the elite" (p. 878). In this sense, education level plays a role—the higher level of education corresponds with a higher level of anti-Americanism (Zimmerman et al. 2020).

The media seem to have played a key role in developing those anti-American feelings. Indeed, Werning Rivera and Bryan (2019) explain that anti-US attitudes appear from the exposure of the public to state-run television that reproduces the anti-Western political landscape in its messaging. This goes in line with the trend proclaimed by Shlapentokh (2011): "popular feelings of xenophobia and anti-Americanism are equally the outcome of positive support for such views on the part of the elite" (p. 879).

Such values and attitudes are also seen as socially desirable, which clearly affects the results of the mass surveys (Shlapentokh 2006). The public shows more apathy towards politics while holding strong anti-American attitudes, which is, as mentioned above, characteristic for hybrid regimes: "To a large degree, this apathy is encouraged by the Kremlin, which gains much more from the passivity of the population than from its political activity, even if it is initially sponsored and directed by the government" (Shlapentokh 2011, p. 883).

These attitudes often take the shape of US-related conspiracy scenarios which may indicate the crisis of trust in government and in the media system. They are flourishing in polarised societies, and simultaneously, they weaken "the state's capacity to govern, and even [lead] to the growth of violent extremism" (Moore 2018, p. 2). One of the narratives that was used in the campaign for the amendments to the Russian constitution in 2020 is that the original Constitution of 1991 was written by the Americans. The Cold War narrative of the bipolar world and the main foreign enemy in the face of the US continues to exist.

As a part of this anti-American trend, Alexei Navalny has been accused multiple times of receiving funding from the US-based organisations, and the roots of "color revolutions" are often attributed to the US influence (see, for example, Polese and Beacháin 2011). One such examples that has been proved to be a conspiracy was described by Hulcoop et al. (2017), who compared the "leaked" email proving the funds from US received by Radio Svoboda with an actual email stolen from the journalist David Satter. Through the minor manipulations with the document, the false impression was created that Navalny (who has never been part of Radio Svoboda investigations team) had been receiving the funding directly from US-based organisations to carry out certain anti-corruption investigations.

The argument that Russian politics is influenced by the US is propagated by the elites, for example, the organisation Nation Liberation Movement (NOD), which has as its goal the preservation of territorial integrity and restoration of Russia's sovereignty from the US. According to their theory, the US has occupied the country after the collapse of the USSR and has several direct agents controlling federal media and the Parliament. The NOD is very active and has representatives in the state Duma (not as an official party).

Anti-Americanism is, therefore, part of the Russian nationalist ideology and a strategy of legitimisation of the state regime based on the juxtaposing of a political and cultural model to the Russian one and by creating an image of Russia as "a besieged fortress, and Putin as the savior of the country" (Shlapentokh 2011, p. 886).

This paper further explores these dynamics within the public sphere that overspills onto YouTube.

### 3. Methodology

This article is part of a broader research program that explores the case of Alexei Navalny challenging the Russian media and political ecosystems; in this case the focus sits on how Navalny is represented on YouTube.

Since its start in 2013, with the motto "the truth is told here", Navalny's activity on his YouTube channels has reached a wide audience. Over the years, his activity has facilitated and embodied the process of extending the actual political debate beyond the margins of the institutional circuit of Russian traditional media and politics towards an online environment outside the traditional media. Navalny does not appear on state television, and he is treated as a "non-systemic" opposition and as a "blogger", not as a politician. In front of this disruptive figure, the Russian government has started a comeback strategy involving astroturfers commenting on YouTube, while leaving the platform relatively free from direct regulatory control and official propaganda.

This paper analyses the YouTube videos mentioning Navalny to enquiry whether and how the hegemonic governmental narratives are present on the platform as a space of communication.

The research question is, therefore, formulated as follows:

*What representations of Navalny are available in the Russian YouTube-sphere?* This RQ has been sub-organised for operational purposes in three aspects:

1. How many videos mention Navalny in the title?
2. What typology of videos is available (Image Type Analysis) and what are their distinctive features (metadata)?
3. In which narratives (plots) has Navalny been incorporated (US-related plots)?

These three steps help identify, nuance, and illustrate some of the aspects mentioned earlier as strategies of the Russian hybrid regime.

The corpus for the analysis was obtained by searching the keyword "Навальный" (Navalny) on the video aggregator tool (Rieder 2015). On 3 September 2020, the search identified around 20,596 YouTube videos associated with this keyword posted between 2006–2020. Those videos were later filtered to include only those in the Russian language containing the word "Navalny" in the title. Also, for relevance and sampling purposes, the videos related to the poisoning case in 2020 were excluded.

The filtered 7449 videos were proportionately structured by videos posted per year and later randomly sampled (95% confidence interval and +− e. 5%). This generated a sampled corpus of analysis of 366 videos distributed according to Table 1. Figure 1 represents the three stages of sampling and distribution of the videos throughout the years.

**Table 1.** Distribution of the videos in the sample.

| Year | Videos Collected with the Keyword "Navalny" | Videos with the Word "Navalny" on the Title | Videos in the Sample |
|---|---|---|---|
| 2006 | 22 | 0 | 0 |
| 2007 | 139 | 2 | 1 |
| 2008 | 254 | 0 | 0 |
| 2009 | 535 | 0 | 0 |
| 2010 | 1352 | 1 | 0 |
| 2011 | 2059 | 236 | 12 |
| 2012 | 1731 | 351 | 17 |
| 2013 | 1572 | 517 | 25 |
| 2014 | 2002 | 211 | 10 |
| 2015 | 1571 | 244 | 12 |
| 2016 | 1450 | 243 | 12 |
| 2017 | 1612 | 1494 | 73 |
| 2018 | 1470 | 1293 | 64 |
| 2019 | 1320 | 1182 | 58 |
| 2020 | 3507 | 1675 | 82 |
| *Total* | *20,596* | *7449* | *366* |

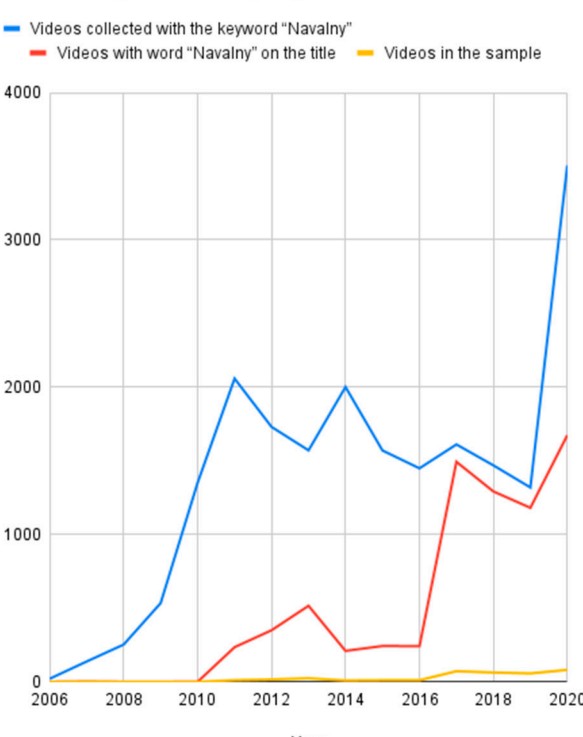

**Figure 1.** Distribution of the videos in the sample in time.

Following the second research sub question, the metadata of the resulting 336 videos in the sample was collected, and the videos analysed following an image type analysis method adapted to videos.

Image type analysis (ITA) "combines quantitative and qualitative features of visual content analysis and allows for the quantification of recurring image types and their qualitative analysis and interpretation" (Brantner et al. 2019, p. 6). Whereas initially, ITA was applied to photography, the technique has successfully been applied to videos (Rodriguez-Amat and Belinskaya 2021). This tool of audio—visual analysis enables the categorisation of big quantities of pictures or videos and treats them as separate groups or types, helping to map the body of images and to identify the dispersion and main tendencies among their topics and approaches. The Image type analysis allows the mapping of the main trends in the corpus and identifies the voices present in the sample and emphasises the structure of the discussion, revealing the groups of discourses and informing the third research question.

The metadata of the sampled videos has been collected: number of views, and average views per type. This operation opens a strand for further explorations.

In the third step of the analysis, the analysis of videos identifies the most common narratives or "plots". This form of content and narrative analysis (Riessman 1993) allows us to distinguish groups of diverse plots framing the figure of Navalny, including the US-related plots and the appeal to anti-American sentiments.

The Results section follows the structure of these three steps. It starts with a descriptive outlook of the collected videos' metadata, such as the number of users, most-watched videos, and the posting dates. Metadata helps illustrate the diversity and the dispersion of the corpus of videos. The metadata is then combined and graphically represented with the types identified in the ITA.

After that, the section describes the available plots visible in the videos representing Navalny as a political opponent. These two forms of analysis, ITA and plots-narrative analysis, are the first steps towards a more exhaustive category map that can be later upscaled and further applied for the study of the representations of Navalny in the Russian

media system as an example of the strategies from the Russian government to secure their own hybrid democratic system. The case, the analysis procedures, and the illustration are also useful for the understanding of the current rhetoric strategies of political opposition used in the political communication in times of digital (and social) media (platforms).

## 4. Results

### 4.1. Metadata and an Early Typology of Videos

The visualisation of the 366 videos led to the identification of several common categories. The bottom-up coding process included a system of three-column labelling on a spreadsheet. The process, based on rounds of analysis and the effort to standardise criteria, identified and grouped a saturated number of repeating distinct categories that identified the following three groups of video-types:

1. Navalny: The videos directly related to Navalny—person and action, include videos posted by Navalny himself, videos about his public speeches, and videos about him as a public figure (in rallies, or as witnesses of fans who filmed him in public (exclusive)).
2. Meta-expressive: The videos that elaborate artistically on Navalny as a figure. This includes artistic, animations, videos and slideshows, and videos that make a satire about political affairs including the sketches performed by Navalny himself. This category also includes the "meta" videos that include posts from bloggers commenting on Navalny or on certain topics (news blogs); it also includes pseudo-documentaries and videos that compile the opinions of people by staging surveys on the street or by phone about the thoughts of the general public about Navalny.
3. Elites: The videos that show typically elite driven discourses, including actual television documentaries and third-party reports; and compilations on Navalny activity. Videos about political communication experts talking about Navalny are also included, as well as videos about political opponents criticising Navalny in YouTube posted interviews.

These three groups of videos organise an initial and more nuanced typology of 11 video-types (see Appendix A).

Figure 2 represents the visualisation and analysis of the 366 videos picked as a proportional sample per year, generating the following distribution of video types as per the typology described above.

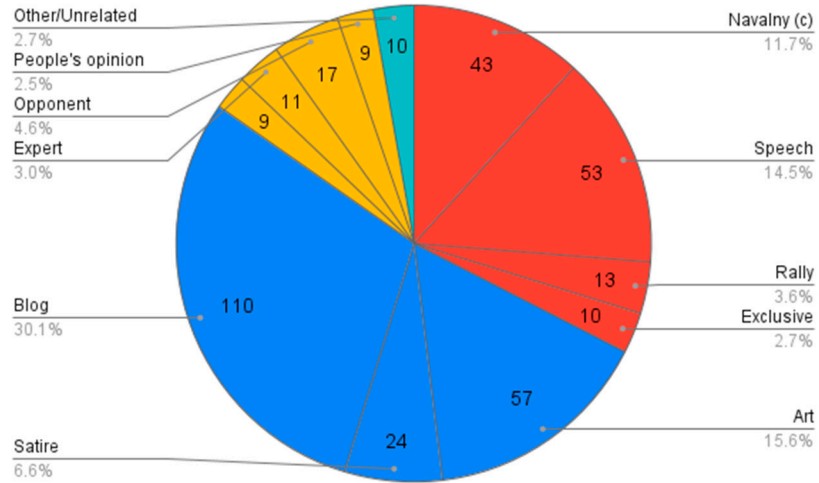

**Figure 2.** Distribution of video types according to image type analysis (*N* = 366).

The three more numerous categories identified include videos posted by netizens individual expression: either as video-blog posts, artistic creations, or the iteration of already existing materials generated by Navalny and his team. In those three occasions (more than 200 videos of the sampled 366), the representations of Navalny are mostly

personal. It can be assumed that most of them echo the most relevant narratives and frames presented by the hegemonic political actors.

The detailed representation of the categories and views can be seen in Table 2.

**Table 2.** Views per category (average).

| Type | Sub-Type | N of Videos | Average Viewers | Max Viewers | Min Viewers | Variation |
|---|---|---|---|---|---|---|
| Navalny | Navalny (c) | 43 | 687.84 | 5056 | 1 | 1114.751488 |
| | Speech | 53 | 4991.36 | 123,200 | 0 | 20,208.99486 |
| | Rally | 13 | 12,565.54 | 147,172 | 45 | 40,476.73309 |
| | Exclusive | 10 | 4988.50 | 42,831 | 0 | 108,314.5577 |
| Meta-expression | Art | 57 | 14,340.81 | 551,391 | 0 | 75,079.99902 |
| | Satire | 24 | 5461.58 | 58,502 | 0 | 13,466.63198 |
| | Blog | 110 | 22,356.15 | 1,465,394 | 0 | 139,633.1351 |
| Elites | TV | 9 | 26,148.89 | 217,993 | 25 | 71,991.29768 |
| | Expert | 11 | 46,379.82 | 391,052 | 9 | 108,314.557 |
| | Opponent | 17 | 29,405.47 | 377,851 | 28 | 90,619.03261 |
| | People's opinions | 9 | 3614.78 | 13,038 | 32 | 4912.989766 |
| | Other/Unrelated | 10 | 152.30 | 675 | 2 | 203.2085061 |
| *Total:* | | *366* | | | | |

The four categories with more views on average (above 20 k) are expert opinion with 46.4 k, opponent opinion (29.4 k views), TV (26.1 k), and blog (22.3 k). These data point out that there is a latent audience in demand for trustworthy sources.

The subsequent narrative analysis is applied only to the second group of videos. This does not include the videos produced by the Navalny team or actual footages from the rallies or speeches, nor the third group which translates the official discourse of pro-Russian government elites.

In 132 videos out of 237 videos (from the second and third types—'meta-expressive' and 'elites') it was possible to identify clear narrative plots. That analysis led to another categorisation of the distinguishable plots. The rest of the videos, however, had not clearly distinguishable plots: either they were merely reporting of events or publications of direct speeches by Navalny. Those were left aside together with other unrelated videos (*N* = 10) that did not have any narrative connection to Navalny and were treated as algorithmic artefacts.

A very early distinction of the 132 videos analysed consisted of checking whether the representation of Navalny was for or against him as a political figure. The results were not disappointing: the amount of activity against Navalny was outstanding.

Figure 3 shows the number of videos containing negative and supportive plots.

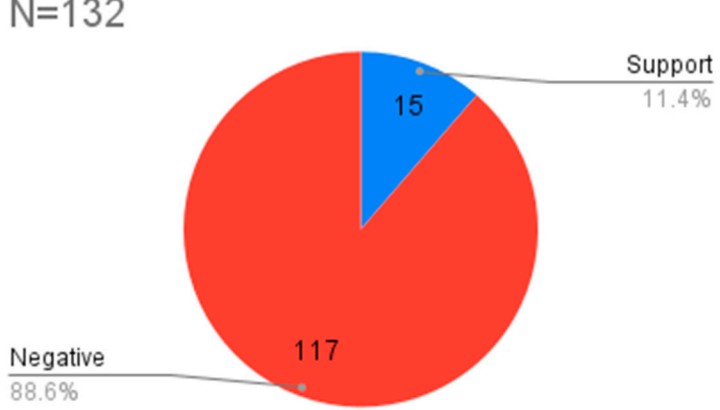

**Figure 3.** Negative vs. supportive videos in the sample (*N* = 132).

The fifteen positive videos expressing support presented Navalny as a victim to the Russian regime, as opinion leader. He was identified as somebody supporting democracy, who wants to make life better, or who helps undermine the current Russian regime.

The 117 videos negatively portraying Navalny are distributed along the years quite regularly, with some peaks clearly visible in 2017 (see Figure 4), when the first big protests were organised by Navalny and the team with the help of YouTube and as a response to a viral investigation on a prime minister and former president Dmitry Medvedev.

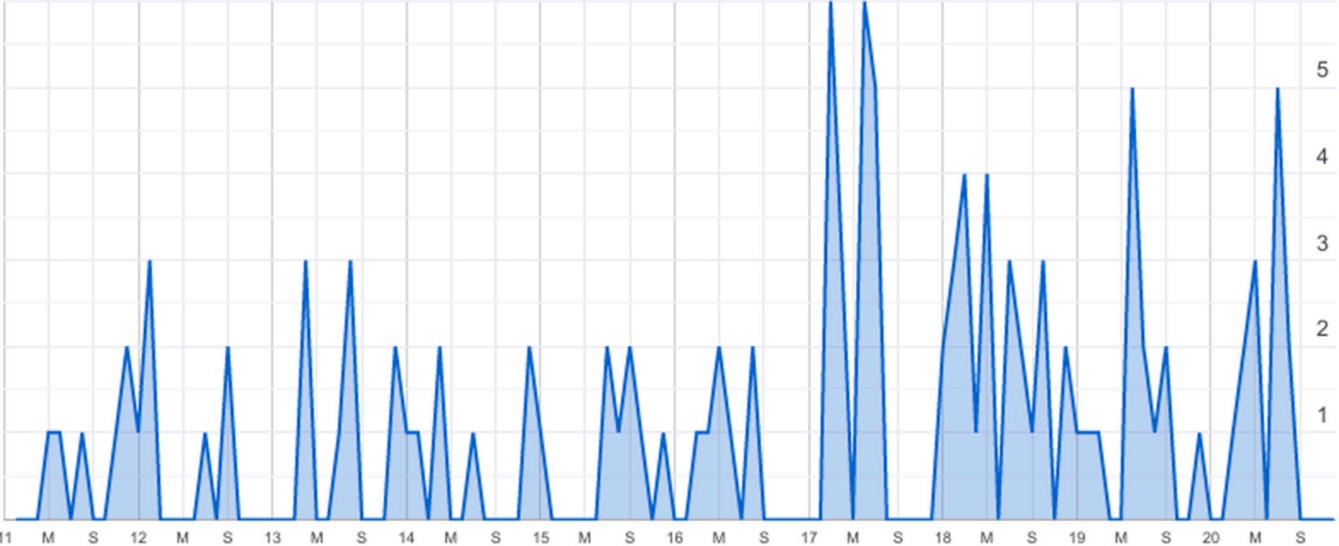

**Figure 4.** Distribution of negative videos in time (between 2011 to 2020).

The negative videos are not particularly rich in terms of audiences (see Table 3). The metadata from the 132 videos analysed showed that there are more than 60% of the videos have less than 1000 views and that they did not stimulate viewers engagement. This is, however, characteristic for the whole category of blogs—there was only one video that had more than 1 million views (1,465,394 views)—news blog (V27) with footage of a famous blogger giving a speech in the Parliament about the cooperation between politicians and bloggers.

**Table 3.** Number of views.

| Views | Number of Videos (Videos with All Scenarios Identified) | Number of Videos (Videos with Negative Scenarios) | Number of Videos (Category "Blogs") |
|---|---|---|---|
| >1 M views | 0 | 0 | 1 |
| >100,000 views | 4 | 4 | 3 |
| >10,000 views | 13 | 12 | 8 |
| >1000 views | 38 | 35 | 33 |
| <1000 views | 77 | 66 | 65 |
| *Total:* | *132* | *117* | *110* |

These videos are powerful in the sense that they tend to build broader narrative plots that situate Navalny at the centre of US-related scenarios. The following chart (Figure 5) shows what stories are repeated. These are what we have called plots.

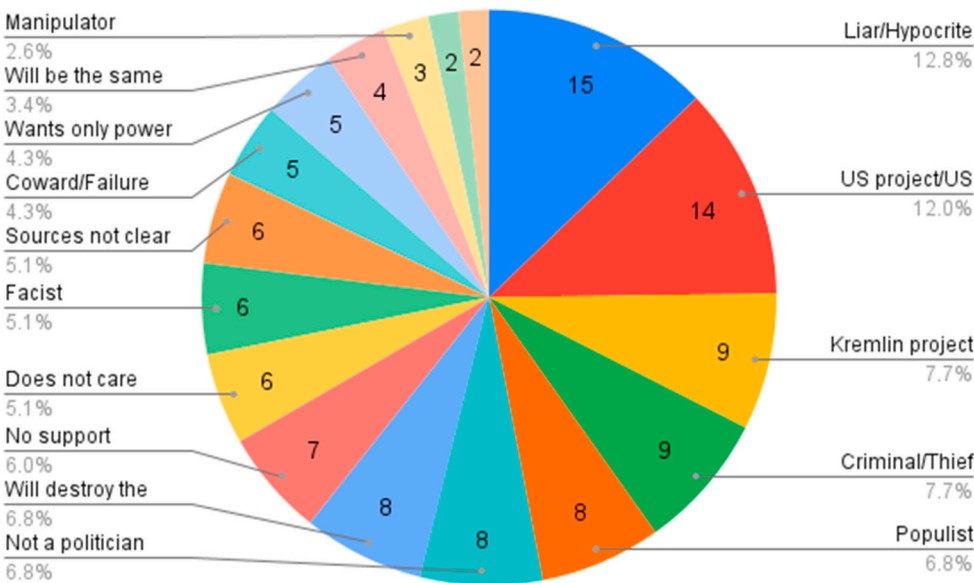

**Figure 5.** Plots distinguished in the 117 negative videos.

*4.2. The US-Related Plots*

Among the videos, Navalny's connection with the US government is highlighted on multiple occasions and in several ways. Here is a description with examples of the usual approaches of this plot:

**The US Connection:**

The US plot is one of the most popular—12% (14) of all videos in the sample discuss possible connections of Navalny to the US. Here, the fact that he studied in Yale is often put forward as an argument. In this example, political scientist Wassermann said:

"I don't believe in the public position of a lawyer. His job is to be a "toilet" or "dumpster"—politicians or businessmen are giving him information in order to compromise the opponents. In Yale he was studying how to overthrow power by nonviolent methods" (V18)[3].

In another example, a blogger is following the meeting Navalny had in Kostroma, presumably with American diplomats, after seeing a car with diplomatic registration plates (V19). Two videos mention, without proof, that Navalny worked for the US embassy. Other bloggers report meetings between Navalny himself or members of his team with English interlocutors.

This "American" scenario does not emerge out of nowhere—it is a popular plot supported by federal channels and various organisations such as NOD. The videos made by NOD are quite frequent on YouTube; in the sample of 366 there were 6 videos by the organisation.

**The US funding:**

Some experts/opponents/bloggers raise questions on Navalny's sources of funding (an expensive office in the centre of Moscow, big team, expensive election campaign); some tend to provide proof of funds tracing back to the US. In one of the videos (V20) the blogger shares proof that in 2006 US National Endowment of Democracy (NOD) transferred 23 thousand dollars to the youth organisation "Da!" (coordinated by M. Gajdar and A. Navalny). In 2007, that same organisation would have transferred 43 thousand dollars, and from 2012, there is no info on funding anymore. In 2015, the organisation was declared illegal on Russian territory. In the video V32, Navalny is accused of having a connection with the US ambassador and described as a mason, which links him with the popular conspiracy theory of a secret mason government.

**The "Kremlin Project":**

This is a second popular scenario, present in many genres: blogs, let's plays, opinions of the experts, and opinions from opponents. This scenario takes multiple shapes, but all

turn around the principle that Navalny is an agent for the Kremlin in charge of revealing or of accessing strategically relevant and convenient information. The multiple arguments that shape that scenario include distracting role, unofficial authorisations, access to privileged information or the favour of the courts, or the leadership of intelligence organisations or even the return of a long-sought debt to the Kremlin.

- Distracting role: in V21 a blogpost about Navalny's reaction to the program "Besogon" in which about microchips inserted via vaccines was discussed. According to the blogger, Navalny only ridicules the host of the program but does not present any actual scientific counter arguments. The blogger concludes that the Kremlin asked Navalny to laugh about the whole theory to stop people from critically assessing the problem.
- Unofficial authorisations: a popular case present in several videos (example: V22) involves flying drones over the residences of prominent figures such as Medvedev—the former President (object of one of the most influential investigations by Navalny). Flying drones is not allowed, which means that he must have received some 'unofficial' and 'secret' permission.
- Access to privileged information: Bloggers often discuss Navalny's sources of information and how some information could have been available to him. The reasonable explanation often voiced in the videos is that information is conveniently leaked to Navalny when a certain politician must be exposed.
- Favorable courts: Even though his brother spent a full sentence (3.5 years) in jail, Alexei Navalny has received only a suspended sentence which was never converted into a real punishment despite his numerous breach violations. He was also allowed to travel, go to the protests, etc.
- Control over youth protests: Navalny is also presented as a project of Russian Federal Security Centre (FSB), a so-called "project youth leader": "And there it is clearly spelled out that it is necessary to take control of the protest youth movement <...> to incept leaders into the youth environment, who, among other things, are allowed to say things more than others, and of course" (V23). The video also discusses the amount of media attention that Navalny receives in case of "an insignificant" protest activity.
- Debt to government: another video presents "factual information" and numbers, discussing the story of Navalny's debt to the government in the size of 4.5 mln rubles, which is a big sum of money for a jobless blogger. After a short period of time his debt decreased to 2.5 million. The author is trying to find an answer as to where one can quickly obtain 2 million (V29). The blogger questions the source of money of Navalny, which is a common narrative among all genres.

All these types coincide with the narrative that the Kremlin "manages" Navalny's work, which makes him only slightly and conveniently uncomfortable for a regime that goes far deeper in the control of the political system. Those perspectives discredit Navalny as a journalist bringing his work ad absurdum and deactivating his activity now turned into another clog of the Russian controlling machine.

Among the videos, another strand of plots consists of treating Navalny as a threat. These are the plots labeled as "Apocalyptic scenarios". Those videos are more projective: they discuss what is going to happen if Navalny comes to power (e.g., wins presidential elections). Again, this scenario takes several shapes: from the threat of a populist regime that could lead to all forms of war and civil unrests to the idea that Navalny is part of a Right-wing organisation with neo-Nazi and authoritarian goals.

- The uneducated Navalny: Navalny's populist agenda shows the lack of understanding of the essence of the political process making his access to political power, a threat that will lead to the full destruction of the Russian Federation V24 (for instance, due to the possible civil war—V25 or the new 'orange' revolution that he is going to organise—V26).
- The ideological Navalny: Another video discloses Navalny's media machinery. It shares that during his campaign for mayor (between July and September 2013), media

belonging to different oligarchs such as Prohorov, Vinokurov, or Mamut in total released 2605 media texts that could only be compared to Hitler or Kim Chen. According to that blogger (V28), all "independent" media in the Russian Federation were working for Navalny contradicting the common belief that Navalny is not equally represented in the media. The comparison with Hitler is also not accidental, in blogs Navalny is often represented as far-right nationalist and compared with authoritarian leaders.

These representations of Navalny as an agent of sorts (either a convenient agent for the Kremlin, or as an agent for the US ideological and geopolitical interests) generate a series of popular videos self-attributed with the role of "debunking myths". These videos often adopt the narrative form of "documentary" (revealing truths about Navalny in style of his own investigations (V20), interviews with the experts (V13)), or the forms of news blog (often generic computer voiceover, no face (V33)), personal political blogs presenting "political analysis" (often the face of the blogger is covered in such videos (V34)), analysis of Navalny videos (revealing lies or inconsistencies in his statements (V35)).

These efforts to present the truth about Navalny extend the efforts of some bloggers presenting "raw facts" and intending to convince the audience with numbers (how much money he received, how much money he owes, how many publications about him have been made). This evidence builds to show how Navalny has exceptional permissions to fly a drone over a secret object which either make him an agent of the Kremlin or an agent of the CIA. These projections of Navalny as a figure extend further towards more transcendental purposes such as that he belongs to the networks of masonry (V32), or even that he has bigger intentions such as the building of a Russian Empire in which he would absolutely reign (or so said the Tarot cards, according to V33).

## 5. Discussion and Conclusions

The number of videos and topics found on YouTube gives clues of how heterogeneous the online discourse on YouTube is. More than twenty thousand videos about Navalny appeared on YouTube between 2006 and 2020. Many of them were directly related to Navalny's activity—through the reposted videos from his own channel, Instagram, speeches at the rallies or at the staff meetings. However, once we dismissed the direct voice of Navalny and focused only on the videos referring to him, a dominant rhetoric emerges: the videos about Navalny work as a mechanism of exclusion that intend to discredit Navalny.

Indeed, the analysis has shown a repertoire of stories that undermine Navalny's views as a critical journalist or regime critic. The plots go, at least, in three directions: one, deactivating his opposition by integrating his work within the general strategic interests of the Kremlin; Navalny is a convenient agent of the Government that pays his debt, or avoids justice, defuses unrests, or chases the right opponent to keep the Kremlin under the same regime. Two, aligning Navalny with an autocratic strategy: with his own agenda for power Navalny aims at taking over the Kremlin with authoritarian, or imperial plans that threaten stability and could start civil wars. Three, connecting his work with the United States global and colonial agenda; Navalny is an agent funded and fed by the American agencies to threaten the sovereignty of the Russian republic. Such an appeal to Navalny's anti-Americanism meets the commonplace of an ingrained Russian nationalist discourse. Scholarship (Shlapentokh 2011; Gudkov 2017; Sokolov et al. 2018; Werning Rivera and Bryan 2019) shows that anti-Americanism is overwhelmingly present and growing across the structure of Russian society. This tendency was also captured by the polls of Levada centre (2021, in Zimmerman et al. 2020): the anti-Americanism, defined there as perception of the US as the main external threat, was steadily growing among all age categories. The US was also chosen as the country with the most "unfriendly" policy in relation to the Russian Federation in all polls made between 2014 and 2020 (Levada 2021).

This research shows that such anti-Americanism is also replicated by the YouTube public. If previous research is right, and anti-Americanism is initially an elitist construct, it can be stated that this discourse is pushed onto YouTube and then re-imagined and

re-activated from within the platform by the public. During more than a decade, this trend was traceable on YouTube, which signifies that the narrative penetrated the YouTube-sphere very early. Thus, without much regulative effort the views from the elites transpire into the public, echoed, accepted and with an appearance of naturality, which returns to legitimise the regime as a whole. The elite-driven anti-American discourse is used as a form of legitimisation for the regime.

YouTube is not an independent platform, and its contents are also not free from interventions, including political and economic pressure. The strength and insistence of pushing discourses into the YouTube-sphere can be a successful strategy for a government to deactivate its challenging opponent. It is only a matter of numbers, of audience, of clicks, of training algorithms to spread the story. And against this strategy YouTube, in its apparent freedom, still serves for governmental consolidation, confirmation, and extension, of, in this case, the slowly built repressed and rotten Russian media ecosystem.

The study has certain limitations due to the very specific character of the study—analysing only videos with the word "Navalny" in the title is not enough to complete the picture of the role of the platform in the full policy struggle. However, this is a sound start that sheds light at least in three directions. First, conceptual: it allows the incorporation of digital media and social media platforms in the constellation of the Russian political communicative sphere, both as a complement and as an extension. Second, methodological, because it sheds some light as per how these extensions and complements could be incorporated in the political communication debates of the Russian regime.

There are more limitations: for instance, the absence of Navalny in the YouTube-sphere. The sample itself builds an artefact that leaves outside videos that consolidate without mentioning the oppositional journalist, and the audience is not considered while it could be a factor to determine the relevance of the videos studied here in comparison to the audience and engagement in other platforms. Since YouTube does not float in the void, this paper should inspire and remind about the urge to research the spread Russian communicative landscape extensively out-reaching towards other powerful and growing platforms: Telegram, TikTok, or Twitter. It is in those platforms where the overspill of the tightened and controlled Russian political sphere is captured and released; and it is there where the challenges to Russian regime can thrive.

It is true that the Russian media ecosystem uses "uncontrolled" platforms like YouTube to flag its "freedom"; while the platform is not entirely free of the political and economic intervention, YouTube has escaped the regulatory waves of the Russian effort, and appears as a tool for challenging and critical disruption. As an unregulated giant driven by algorithmic recommendation bubbles the sole form of intervention from government seems to rely on the long tentacles of multiplied and spread convenient content-driven activities. The anti-American discourses capturing the figure of Alexei Navalny are designed to penetrate the territories of the emotion, and the storytelling strategies that mix fiction, news, and Tarot to deactivate the inconvenient critical work. All these are tools that add and further exclude Navalny as an acceptable political actor who has been already ghosted from the official stage.

This effort to control the online environment of the social media platforms is one of the dilemmas of hybrid regimes (Petrov et al. 2014) and the dictator's dilemma (Shirky 2011) and in this case, describes the paradox within which the current state of the Russian media system finds itself well. There is a general and slow co-optation of the digital media environment and trends of tightening the media and digital policies, and, as this paper argues, the exceptionality of YouTube confirms the hybridity rule: hybrid regimes tolerate some amounts of oppositional media to maintain the legitimacy, and the Alphabet-owned video platform which falls beyond the reach of the legislation aimed at traditional and Russian-based media enjoys some American-made freedom. YouTube can be "intervened from within" with astroturfing. But without the presence of ground-truth data (like with leaked lists of astroturfers, see Belinskaya 2020, for this), it does not matter if the videos analysed here were prompted by the Kremlin-affiliated bloggers. Instead, this paper

illuminates the strategic use of stories and available narratives within the platform. It is a particular rhetoric that extensively contributes to the fake operations of legitimacy practiced by the Russian regime.

There is no consensus on the nature of the Russian political regime. The vague idea that it is a hybrid regime in a transitional stage from an authoritarian to a democratic regime is difficult to hold nothing seems to lead towards stabilisation or towards the strengthening of the democratic institutions but rather the contrary. A systematic effort of financial and judicial pressure, combined with recent attacks on independent media outlets and on news aggregators, show a zealous effort to increase control over the digital informational environment and seem to consolidate an overall trend that extends with smart moves towards the social media platforms, only to consolidate a path towards (neo)authoritarianism.

**Funding:** This research received no external funding.

**Data Availability Statement:** The initial dataset can be made accessible by demand.

**Acknowledgments:** I would like to express my special thanks to my colleague and friend Joan Ramon Rodriguez-Amat for his support and feedback. Open Access Funding was provided by the University of Vienna.

**Conflicts of Interest:** The author declares no conflict of interest.

## Appendix A

**Table A1.** Nuanced typology of the videos.

| | Type | N of Videos |
|---|---|---|
| | **Group 1: Navalny** | |
| 1 | Navalny (c)—videos reposted without any change from Navalny's two Youtube channels or his Instagram page. These videos had more informational character and did not represent the opinion of the author who posted the video (for example, Navalny commenting about some incident in his weekly blog). | 43 |
| | Speech is the category where Navalny himself gave a speech and was shot (presumably) by the author of the video at the rallies, meetings with staff in the regions, court sittings, office meetings, or interviews of Navalny given to journalists or popular bloggers | 53 |
| | Rally—shootings from rallies, images of the crowds with Navalny not present. | 13 |
| | Exclusive—"hidden cam" shooting or videos when authors, for example, spotted Navalny in their cities. | 10 |
| | *Total videos* | *119* |
| | **Group 2: Meta-expressive** | |
| 2 | Art included all possible forms of personal expression that did not fall under category blog or satire—such as animations, music videos, songs, (funny/artistic) compilations, slide shows, let's plays. | 57 |
| | Satire included various forms of political humour—sketches, satirical proofs of conspiracy theories, jokes, anecdotes, parodies, Navalny's own sketches. | 24 |
| | Blog—original videos representing the opinion of the blogger on a certain topic or narration of the events ("news blogs") by the blogger who can be visible in the video or not. | 110 |
| | Peoples opinion—surveys on the streets or on the phone, "general public" is giving an opinion about Navalny. | 9 |
| | *Total videos* | *200* |

**Table A1.** *Cont.*

| | Type | N of Videos |
|---|---|---|
| | **Group 3: Elites** | |
| 3 | TV—reposts of the actual TV reports about rallies or other news concerning Navalny, investigations and other materials shown on TV. | **9** |
| | Expert opinion category included the opinion of the experts, mostly political scientists. | **11** |
| | Opinion of the opponent—in most of the cases parts of the interviews of the political figures (for example, other candidates for mayor or presidency) answering questions about Navalny. | **17** |
| | *Total videos* | *37* |
| 4 | Other (Unrelated) | **10** |

## Notes

[1] The Federal Service for Supervision of Communications, Information Technology and Mass Media—the tasks include supervision in the field of communication and media, as well as supervision of personal data protection and radio frequency service organisation activities.

[2] Or so-called 'Yarovaya package'—a set of amendments to the Federal laws containing proposals to fight extremism and terrorism online ("public justification of terrorist acts"). Passed in 2016 (Available online: https://sozd.duma.gov.ru/bill/1076089-7 acceded on 29 October 2021).

[3] For ethical compliance the specific references to videos are not published; the author is happy to make the database accessible upon request.

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
