# Peer review of "The Ghosts Navalny Met: Russian YouTube-Sphere in Check"

_journalmedia, doi:10.3390/journalmedia2040040_

Round 1

Reviewer 1 Report

the article is well written and well set up. THE presentation of the literature is accurate and perhaps far too detailed compared to the article as a whole.
It would be necessary to enrich the description of the results of the research and to make the conclusions less elementary. Interesting, but could be better argued

Reviewer 2 Report

This article is very interesting. Based on an extensive corpus of Russian YouTube videos that refer to Navalny, the author shows through an Image Type Analysis that the platform is replete with “narrative plots that situate Navalny at the centre of conspirational stories”, in congruence with the legitimization strategies of the Russian regime.

However, the manuscript should be reread carefully in order to eliminate some formal problems. It would be necessary, for example, to correct “However, a decade later However recently Russia” (p. 7, line 323), to add references when the author states that “the roots of “color revolutions” are often attributed to the US influence” (p. 10), or to revise the sentence “The US influence on Russian politics is propagated by [Russian] elites” (p. 10). (It is not the US influence on Russian politics that is propagated by elites, but rather the argument of the US influence on Russian politics that is propagated by [Russian] elites”).

In addition, some assertions would need more elaboration (for example when the author states that “YouTube is not a free platform; and its contents are also not free”, p. 18, line 828).

Reviewer 3 Report

This was a potentially valuable contribution to knowledge on contemporary political communication in Russia on the digital sphere. However, the apparent lack of objectivity and one-sided interpretations have excluded this outcome. Here are some of the observations and suggestions, appearing in no particular order.

1) The control of Russian mass media occurred before Putin's presidency, in fact this was being done systematically from at least 1998 by Yeltsin. 

2) Why capital F in Freedom on page 3, line 133?

3) Freedom House is not a reliable academic source as it is a State Department public diplomacy asset and given the US classifies Russia as an "enemy state" in the context of the New Cold War makes it as dubious as a source as Wikipedia. 

4) Page 2 line 152 - You do know who Dorenko was? The 'TV Killer' of the 1990's and one of the biggest character assassins hardly an 'independent' or 'good' journalist. 

5) There is a very narrow selection and reliance on secondary academic sources on the Russia mediascape, many big names are missing from here.

6) The top line on page 4 is pure and utter conjecture and speculation that is really on the line of what is legally acceptable let alone ethically, it is not proven. 

7) There is a lot of conjecture on Kremlin control of journalists and media outlets, however recent research indicates that this is more active at the local and regional levels of government too (see Erzikova and Lowrey 2020 for example).

8) As stated in the manuscript, YouTube (and social media in general) are not neutral or non-partisan communication platforms, their actions and responses during COVID-19 have demonstrated this as a fact. Also Morozov stated as such in his dark side of the web.

9) There is a very heavy reliance on an extremely narrow set of academic sources. 

10) The logic of the paper is not consistent owing to the various applications of concepts and political symbolism - "Russian dictator" but admitting to the system as being "hybrid" and not "authoritarian." 

11) The use of vague slogans as "anti-Americanism" without any real attempts to understand (or admit to) the contextual situation is a meaningless and pointless exercise for any chance of objective knowledge seeking and building. Being critical of US policy does not make one automatically "anti-American." 

12) The government's issue is not only with US contacts, but also UK as well. After all, Navalny holds a UK passport (and therefore cannot run for the State Duma or presidency). The recent meeting of one of the key Navalny team members in Moscow with an intelligence official from the UK embassy actually makes a case for their concerns.

13) The author(s) seem to create a web of conspiracy theories about alleged conspiracy theories as the logic and strength of argument is paper thin. 

14) The wording of the research question is not very academic and should be rephrased in a clearer and more theoretically meaningful manner. 

15) Navalny is not what one would call a Western oriented democrat, based on his own words. He refers to Central Asians as being sub-humans that should be deported, and liberals as cockroaches. 

16) The conclusion is very weak. These questions should be also addressed. What is the reliability of the study? What are the study's weaknesses? Are the results generalisable or not? What is the possible research agenda that stems from this manuscript? 

Round 2

Reviewer 3 Report

The revised manuscript is a marked improvement upon the original one, however, there are still some minor issues that remain to be dealt with. 

1) There is a typo on page 3, McQuail and not MaQuail

2) The task of capturing the media assets owned or controlled by the oligarchs, such as Berezovsky and Gusinsky, was a very easy one. Under Yeltsin they had managed to gain access to state bank loans at very favourable terms owing to their 'loyalty' to Yeltsin in the wake of the 1996 presidential elections. It was merely a matter of calling these in as media assets were intended for informational persuasion and influence and not as profit making enterprises. This should certainly be reflected in explaining the demise of private media assets.

3) On page 19 there is a mention "that anti-Americanism is overwhelmingly present and growing in Russian society." In ALL of society? In SEGMENTS of society? What are the demographics, where and when were the polls? This is a glittering generality and really needs to be solidly supported and referenced. It also depends completely on how the term "anti-Americanism" is defined and explained. 
